# Repair Mechanisms of the Neurovascular Unit after Ischemic Stroke with a Focus on VEGF

**DOI:** 10.3390/ijms22168543

**Published:** 2021-08-09

**Authors:** Sunhong Moon, Mi-Sook Chang, Seong-Ho Koh, Yoon Kyung Choi

**Affiliations:** 1Department of Bioscience and Biotechnology, Bio/Molecular Informatics Center, Konkuk University, Seoul 05029, Korea; msh9504@konkuk.ac.kr; 2Department of Oral Anatomy, Seoul National University School of Dentistry, Seoul 03080, Korea; mschang@snu.ac.kr; 3Department of Neurology, Hanyang University Guri Hospital, Guri 11923, Korea; ksh213@hanyang.ac.kr

**Keywords:** stroke, vascular endothelial growth factor (VEGF), blood–brain barrier (BBB), stem cell, gaseous molecule, regeneration

## Abstract

The functional neural circuits are partially repaired after an ischemic stroke in the central nervous system (CNS). In the CNS, neurovascular units, including neurons, endothelial cells, astrocytes, pericytes, microglia, and oligodendrocytes maintain homeostasis; however, these cellular networks are damaged after an ischemic stroke. The present review discusses the repair potential of stem cells (i.e., mesenchymal stem cells, endothelial precursor cells, and neural stem cells) and gaseous molecules (i.e., nitric oxide and carbon monoxide) with respect to neuroprotection in the acute phase and regeneration in the late phase after an ischemic stroke. Commonly shared molecular mechanisms in the neurovascular unit are associated with the vascular endothelial growth factor (VEGF) and its related factors. Stem cells and gaseous molecules may exert therapeutic effects by diminishing VEGF-mediated vascular leakage and facilitating VEGF-mediated regenerative capacity. This review presents an in-depth discussion of the regeneration ability by which endogenous neural stem cells and endothelial cells produce neurons and vessels capable of replacing injured neurons and vessels in the CNS.

## 1. Overview of Repair Mechanisms Following Brain Damage

Ischemic stroke, which accounts for 87% of all stroke cases, results from a sudden cessation of adequate amounts of blood supply to parts of the brain. The vascular system plays a critical role in supplying oxygen (O_2_) and nutrients to neuronal systems. An ischemic stroke typically presents with a rapid onset of neurological deficit. Cell–cell communication in the neurovascular unit contributes to a functional neurovascular system through an orchestrated network of extracellular matrix, endothelial cells (ECs), pericytes, astrocytes, oligodendrocytes, microglia, neural stem cells, and neurons. Thus, interruption of blood flow through an intracranial artery leads to deprivation of O_2_ and nutrients to the vascular territory, resulting in metabolic changes in the surrounding cells, such as abnormal mitochondrial activity, inflammation, disruption of the blood–brain barrier (BBB), and cell death. Functional recovery after an ischemic stroke may depend on the fate of the ischemic penumbra if the circulation is re-established in time. If not, at the onset of a stroke, the complex and dynamic association between the brain vasculature and neuronal system delays functional recovery.

Elucidating the mechanism of neurovascular repair is important to develop therapeutic strategies to attempt to reverse or minimize the effects and to prevent future infarcts. Clinical trials aiming to develop strategies for neuroprotection with respect to ischemic stroke have failed to demonstrate clinical efficacy, possibly due to limited regenerative capacity in the central nervous system (CNS) [1]. In this review, we specifically discuss the repair mechanisms related to neurovascular protection and regeneration through stem cell-mediated repair mechanisms and gaseous molecule-mediated regenerative signaling. Additionally, strategies for overcoming a stroke are also discussed, with a specific focus on cellular therapy and molecular mechanisms involving vascular endothelial growth factor (VEGF). The VEGF family includes VEGF-A, -B, -C, -D, -E, and placental growth factor, which bind in a distinct pattern to three structurally related receptor-type tyrosine kinases, namely, VEGF receptor 1 (VEGFR1), VEGFR2, and VEGFR3. VEGF-A is a major growth factor that binds to VEGFR1 and VEGFR2 but not to VEGFR3 [2]. In this review, we have focused on VEGF-A and refer to it as VEGF because VEGF acts on entire neurovascular cells such as endothelial cells, neurons, and glia [3,4,5,6,7,8]. All these mechanisms involving stem cells and gaseous molecules partly share common molecular pathways and signaling molecules such as VEGF and its related factors.

## 2. Mechanisms Associated with the Neurovascular Unit Underlying Acute and Chronic Stroke

### 2.1. Neurovascular Damage Due to Acute and Chronic Stroke

Ischemic stroke can be divided into acute or chronic based on the time from onset. Ischemic stroke is classified as hyperacute, acute, subacute, and chronic with an onset time of 0–24 h, 1–7 days, and 1–3 weeks, and more than 3 weeks, respectively. ECs play key roles in neuroprotection by maintaining the integrity of the BBB during a stroke [9]. ECs in the CNS possess tight junctions (occludin, claudin, zonula occludens-1 (ZO-1), ZO-2, and ZO-3) and adherence junctions (VE-cadherin), which are reliant on several interdependent mechanisms (ionic dysregulation, inflammation, oxidative and nitrosative stress, enzymatic activity, and angiogenesis) for their functions [10]. In the neurovascular unit, surrounding cells, such as astrocytes and pericytes, contribute to the formation and maintenance of the BBB [11,12] by downregulating the expression of vascular permeability factors such as VEGF [13,14].

Strbian D. et al. demonstrated monophasic BBB leakage, starting as early as 25 min (acute phase) after post-ischemic reperfusion, and lasting for 3 weeks (chronic phase) detected by Evans blue staining [15]. In stroke patients, VEGF levels are increased in the neurons, astrocytes, and ECs of the ischemic penumbra [16]. In the acute phase of a stroke, excessive VEGF acts as a potent permeability factor [17,18]; however, in the chronic phase of ischemic injuries, administration of VEGF may strongly induce regenerative signaling, thereby mediating angiogenesis, neurogenesis, and synaptic function [4,5,6,19,20,21,22]. Thus, we discuss the neurovascular functions after a stroke focusing on cellular and molecular mechanisms associated with VEGF-mediated dual functions during the acute and chronic phases of a stroke.

#### 2.1.1. Neurovascular Damage within Hours

A stroke directly damages the neural tissue and induces tissue ischemia attributed to vascular occlusion. During an acute stroke, ischemia/reperfusion leads to the disruption of BBB, loss of pericytes, activation of glia, macrophage infiltration, immune activation, inflammation, and neuronal cell death [10,23]. Pericyte constriction due to oxidative–nitrative stress has been observed 2 h after ischemia–reperfusion in the mouse brain [24]. Pericyte contraction entraps erythrocytes at the capillary constriction sites during a stroke, thereby obstructing microcirculation [24,25]. Occlusion of the cerebral artery leads to focal ischemia and upregulation of VEGF expression as early as 2–6 h after vessel occlusion [26,27]. VEGF levels are normalized within 12 h, are increased again following 3–7 days after reperfusion, and subsequently restored to baseline levels after 2 weeks [28]. Excessive VEGF levels demonstrate deleterious consequences such as BBB breakdown, vascular leakage, and brain edema [14,18]. Activation of VEGF/VEGFR2 signaling throughout stroke progression results in harmful effects during the acute phase of ischemia–reperfusion. The binding of VEGF to VEGFR2 stimulates vascular leakage through downstream signaling pathways, including the phosphoinositol-3-kinase (PI3K)-protein kinase B (PKB; Akt)-endothelial nitric oxide synthase (eNOS) axis in ECs [29]. Hypoxic astrocyte-derived VEGF binds to VEGFR2 expressed on human retina microvascular ECs, leading to HIF-1α gene expression and consequent VEGF upregulation [30]. Inhibition of VEGFR2 signaling in ischemic stroke has been linked to a reduction in edema and infarct volume [31,32].

Impaired NO-mediated vascular functions are observed when NO reacts with reactive oxygen species (ROS) in the core region of the damaged brain. These effects can be mitigated by superoxide dismutase [33]. NADPH oxidase 2 (Nox2) may be the source of ROS since Nox2-deficient mice fail to demonstrate vascular tone after ischemia [33]. Reactive nitrogen species (RNS)-mediated structural alterations of vascular endothelial cadherin (VE-cadherin), attributed to nitrosylation of serine residues on VE-cadherin in ECs, are regarded as the mechanisms underlying early leakage in BBB [34]. In addition, the Src–Rac1–PAK (p21-activated kinase) pathway phosphorylates VE-cadherin, followed by its dissociation [35]. Furthermore, ischemic neurons facilitate astrocyte-derived VEGF production, leading to BBB breakdown via a reduction in the levels of tight junction proteins (occludin and claudin-5) [22] (Figure 1).

Additionally, astrocyte and microglial activation may occur immediately after a stroke, amplified by multiple inflammatory factors, including tumor necrosis factor-α (TNF-α), interferon-γ (IFN-γ), interleukin-1β (IL-1β), IL-6, and adenosine triphosphate (ATP) [36,37]. These molecules act on a variety of recognition receptors expressed by microglia, astrocytes, oligodendrocytes, and oligodendrocyte precursor cells (OPCs) and trigger the activation of these cells [38,39]. Morphological changes in glial cells have been detected at varying distances and depths from the ischemic lesions in the ischemic brain [37,40].

#### 2.1.2. Neurovascular Damage within Days

Following 1–2 days after injury, weakened tight junctions attract the infiltration of immune cells into ischemic brain tissues. Peripheral monocytes attach to the damaged brain ECs expressing intercellular adhesion molecule-1 (ICAM-1) and vascular cell adhesion molecule-1 (VCAM-1), which are the downstream factors of the kappa-light-chain-enhancer of activated B cells (NF-κB) observed during acute CNS inflammation [41] (Figure 1).

Inflammatory conditions lead to the downregulation of eNOS expression in ECs [42]. eNOS-derived NO possesses vasodilatory and anti-inflammatory properties as well as stroke-protective properties [43]. Neutrophils and other monocytes rapidly infiltrate the brain tissue and release additional pro-inflammatory cytokines that elicit a more robust reactive glial response and reinforce local inflammation [44]. In the ischemic core, round and amoeboid microglia can be observed at 1–3 days after a stroke. This effect stems from the changes in the morphology and polarization of microglia [40]. Some reactive astrocytes rapidly proliferate and densely populate the area around the lesion core within the 7–10-day period of glial scar formation [45,46]. Glial scars inhibit excessive inflammatory responses and restrict cellular degeneration [47].

The beneficial effects of VEGF in the subchronic phase of ischemia–reperfusion include tissue oxygenation via promotion of vasodilation, arteriogenesis in the penumbra area, angiogenesis, mitochondrial biogenesis, and neurogenesis [19,20,26,48]. VEGF improves neurological recovery after 48 h but not 1 h after ischemia [18]. In a rat model of transient focal ischemia induced by middle cerebral artery occlusion (MCAO), VEGF (1–3 days after ischemia) exerts multiple effects, such as neuroprotection, neurogenesis in the dentate gyrus and the subventricular zone, and angiogenesis in the striatal ischemic penumbra [48]. The in vivo ischemic brain injury model exhibited enhanced HIF-1α immunoreactivity in the peri-infarct region of the wild-type mice, which was abolished in heme oxygenase (HO)-1^+/−^ heterozygote knockout (KO) mice [49]. HO-1-derived carbon monoxide (CO) stabilized HIF-1α via inactivation of prolyl-4-hydroxylase domain 2 (PHD2) due to O_2_ depletion, and HIF-1α induced ERRα expression [49]. PHD2 deficiency in neurons stabilizes HIF-1α protein and consequent VEGF mRNA levels not only in the neurons but also in astrocytes, demonstrating neuron-to-astrocyte signaling and pro-angiogenic responses [50]. Neural stem cells (NSCs) are self-renewing cells that can generate neurons, astrocytes, and oligodendrocytes. VEGF-overexpressing transgenic mice exhibited enhanced NSC proliferation, migration, and survival in the MCAO model [21]. As VEGF does not cross the BBB, circulating VEGF may activate neurogenesis indirectly in the CNS by upregulating the expression of neurotrophic factors, such as brain-derived neurotrophic factor (BDNF) in ECs [51,52].

#### 2.1.3. Neurovascular Damage within Weeks

Immune-mediated inflammatory responses gradually cease within weeks after the injury, followed by endogenous repair. Upon endogenous angiogenesis and neurogenesis, increased cerebral blood flow (CBF) and maturation of ECs (tight junction integrity and interaction with pericytes) contribute, in part, to functional neural circuits.

In this phase, the glial scar matures completely partly via the signal transducer and activator of the transcription 3 (STAT3)-dependent mechanism [45]. In the STAT3-KO mice, the elongated processes of scar-forming astrocytes remain perpendicular to the lesion core and do not turn parallel to form a dense astrocytic scar border after CNS injury [45]. Astrocytes within the glial scar produce molecules that inhibit axonal growth. Semaphorin 3A, several ephrins, slits, and matrix components inhibit axon growth within the glial scar [46]. However, recent studies suggest that glial scarring stimulates regeneration [53]. STAT3 binds to the VEGF promoter, improves VEGF mRNA and protein expression, and enhances angiogenesis after a stroke [54]. In ischemic white matter injury, STAT3 is also involved in inducing morphological changes in the microglia (demonstrating an anti-inflammatory phenotype) mediated by the immune modulator, fingolimod [55].

In the next section, we will discuss regeneration mechanisms in the neurovascular unit after a stroke, specifically with respect to angiogenesis, gliogenesis, neurogenesis, and synaptogenesis.

### 2.2. Regeneration

In humans, functional repair following a severe stroke is very difficult [56]. Researchers have attempted to decipher the precise cellular and molecular mechanisms to overcome the limited ability of neurovascular recovery leading to normal brain functions after a stroke [1,57]. One critical aspect associated with the poor regeneration of central axons and glia is the inability of the environment to support axonal growth and myelination [57]. The neurovascular network may amplify the repair signals, and then support axonal regrowth by enhancing endogenous angiogenesis, gliogenesis, neurogenesis, and new synapse connections [58]. In this section, we discuss VEGF-mediated signaling pathways with respect to neurovascular components, such as neurons, NSCs, ECs, pericytes, astrocytes, microglia, and oligodendrocytes.

#### 2.2.1. Neurons/NSCs

Accumulating evidence indicates that the VEGF–VEGFR2 axis mediates key neurobiological processes involved in neurogenesis, hippocampal plasticity, learning, and memory [3,20]. Co-activation of the NMDA receptor (NMDAR) and VEGFR2 in hippocampal pyramidal neurons triggers synaptogenesis and promotes synaptic targeting of NMDAR and VEGFR2 [4]. Overexpression of VEGF in vivo by intracerebral administration, gene transfer, or in conditional transgenic animal models demonstrates a VEGFR2-mediated increase in adult neurogenesis and improves hippocampal-dependent cognition [59]. Moreover, VEGFR2 is required for the beneficial effects of anti-depressant therapies on fear-related behavior [4]. In contrast, the depletion of endogenous VEGF via small hairpin RNA silencing or inducible expression of a VEGF trap leads to altered hippocampal neurogenesis in response to an enriched environment [3,6] or selective deficits in memory [59]. Both VEGF and VEGFR2 are expressed in the pyramidal cell layer and proximal apical dendrites in the cornu ammonis 1 (CA1) and CA3 regions of the hippocampi in postnatal day 15 mice [4], indicating that hippocampal pyramidal cells can respond to endogenous VEGF. Treatment with an inhibitor that specifically inhibits the tyrosine kinase activity of VEGFR2 diminishes the amplifying effect of VEGF [4]. NSCs, genetically modified by deleting the VEGFR2 gene, inhibits synaptogenesis, which is identified by detecting the expression of the presynaptic vesicle marker synapsin-1 and/or the postsynaptic density protein-95 [4]. Conditional KO of VEGFR2 via the Nestin-Cre system triggers selective VEGFR2 deletion in neural cells. Hippocampal neurons exhibit increased NMDA-type glutamate receptor (GluNRs)-mediated synaptic responses when VEGF is administered, whereas VEGF fails to induce any significant change in GluNR excitatory postsynaptic currents in VEGFR2 conditional KO slices [4]. Therefore, NSC-derived VEGFR2–VEGF signaling may trigger regeneration processes such as neurogenesis and synaptogenesis, leading to enhanced long-term memory.

#### 2.2.2. Endothelial Cells (ECs)

Endogenous VEGF produced by ECs is crucial for vascular homeostasis [60]. Specific deletion of VEGF in ECs using VE-cadherin-driven Cre recombinase showed signs of severe failure of the cardiovascular system such as hemorrhages, microinfarcts, EC rupture, and vascular constriction [60]. In this study, exogenous VEGF was unable to compensate for the loss of endogenous VEGF in ECs despite the fact that signaling through VEGFR2 activates several downstream pathways, including the PI3K–Akt pathway, which is responsible for cell survival. Micro-hemorrhages observed in VEGF-deficient ECs in adult mice (VEGF^ECKO^) are likely a reflection of the focal disruption of the vasculature attributed to EC death, as observed in vitro [61]. Additionally, the sudden death of a significant number of mutant VEGF^ECKO^ mice was also observed [61]. In human umbilical vein ECs (HUVECs), the inactivation of VEGF results in mitochondrial fragmentation (as revealed by Hsp60 immunocytochemistry), the suppression of cell metabolism, and autophagic cell death [5]. These cellular phenotypes are mediated by forkhead box protein O1 (FOXO1), which is robustly stabilized in the absence of VEGF. When FOXO1 is deleted from VEGF-deficient endothelium, autophagic cell death is rescued, suggesting that the main role of autocrine VEGF is to regulate FOXO1 levels to maintain cellular homeostasis [5]. Therefore, the maintenance of baseline levels of VEGF in the endothelium is required to support vascular homeostasis.

Activation of VEGFR2 in ECs via binding with a strong agonist, VEGF, results in Akt-dependent phosphorylation of eNOS as well as NO production [62] (Figure 1). Increased CBF attributed to the VEGF–eNOS-NO pathway may facilitate ischemic tissue repair after a stroke [43]. Intravenous treatment with the eNOS substrate l-arginine mediated the blood-flow-dependent recovery in a rat model of an experimental stroke [63]. NO can increase the number of circulating endothelial progenitor cells (EPCs) after a stroke, consequently triggering neovascularization [64,65]. Treatment of human microvascular ECs with recombinant VEGF increases protein kinase C-mediated HIF-1α gene expression, leading to increased expression of downstream target genes such as glucose transporter-1 and VEGF [30]. Therefore, secreted VEGF influences vascular tone, angiogenesis, and glucose transport in the vascular system (Figure 1).

#### 2.2.3. Astrocytes

Neuronal survival relies in part on astrocytic antioxidant abilities [66,67,68]. Following a stroke, ischemic tissues demonstrate O_2_ and energy depletion, and subsequently, acidosis, inflammation, glutamate excitotoxicity, and ROS/RNS generation [69]. Oxidative stress leads to the fragmentation and autophagic degradation of mitochondria through excessive Ca^2+^ influx in rat astrocytes [70]. Transfer of mitochondria from healthy astrocytes to adjacent ischemic-damaged neurons enhances neuronal survival and improves functional behavioral outcomes [71].

In astrocytes, Ca^2+^ signaling stabilizes the hypoxia-inducible factor-1α (HIF-1α), thereby inducing the expression and secretion of VEGF [19,49]. Upon CNS injury, HIF-1α, a major transcription factor for VEGF, may trigger regenerative signaling pathways by manipulating various transcription factors [72,73]. The HO-1–CO pathway-mediated entry of extracellular Ca^2+^ occurs through L-type voltage-gated Ca^2+^ channels (VGCCs) [19]. Ca^2+^ influx into astrocytes activates calmodulin-dependent protein kinase ββ (CaMKKββ)-mediated expression of AMP-activated protein kinase α (AMPKα) by enhancing phosphorylation and nicotinamide phosphoribosyltransferase (NAMPT)-mediated sirtuin 1 (SIRT1) activation. Subsequently, the peroxisome proliferator-activated receptor γ-coactivator-1α (PGC-1α)-estrogen-related receptor α (ERRα) axis is activated. These events result in upregulated VEGF expression and secretion [19]. The L-type VGCC-mediated PGC-1α-ERRα axis strongly induces human astrocytic mitochondrial biogenesis [74]. Consequently, an increase in O_2_ consumption stabilizes HIF-1α, leading to VEGF secretion and ERRα expression [49,75]. ERRα, a transcription factor for VEGF, also binds to HIF-1α and PGC-1α, leading to enhanced VEGF transcription [19,49]. Taken together, L-type VGCCs in human astrocytes may be involved in angiogenesis and mitochondrial biogenesis (Figure 2).

In primary rat cortical astrocytes with reduced O_2_ availability (hypoxia and anoxia), the inhibition of protein kinase A reduced the expression of HIF-1α and phosphorylation of the cyclic AMP-responsive element-binding protein (CREB) and induced cell death [76]. Astrocyte-derived VEGF also activates the eNOS–NO pathway [77], possibly leading to the regulation of the vascular tone in ECs [78] (Figure 2).

#### 2.2.4. Pericytes

As pericytes are crucial for the maintenance of BBB integrity, they can also affect stroke progression and recovery. Brain pericytes are derived from neural crest cells and demonstrate various functions, including CBF regulation and BBB maintenance by communicating with other cells, especially ECs and astrocytes. The ratio of brain pericytes to ECs can range from 1:1 to 3:1 (endothelial-to-pericyte) [79,80]. Pericyte–EC interactions are prominent for BBB maintenance with profound effects on the basement membrane and endothelial tight junction architecture and function. During a stroke, pericyte detachment from ECs and pericyte death can influence BBB permeability [81]. The contractile properties of pericytes provide the capacity to regulate capillary blood flow. However, this may exert detrimental effects on oxidative–nitrative stress [24]. ATP and noradrenaline induce capillary constriction mediated by pericytes [82]. Glutamate reverses noradrenaline-induced constriction via prostaglandin E_2_ and NO [25,82].

Platelet-derived growth factor (PDGF), VEGF, transforming growth factor-β, angiopoietin-1, and Wnt pathways play critical roles in EC–pericyte interaction [80]. Pericyte-derived angiopoietin-1 upregulates the expression of tight junction proteins in endothelial cells via the Tie-2 receptor [83]. EC-derived PDGF attracts pericytes. In addition, pericyte-derived angiopoietin-1 attracts Tie-2 receptor-expressing ECs [84]. These interactions mediate pericyte attachment and coverage of newly formed blood vessels, resulting in the maintenance of vascular homeostasis and maturation. This interaction can be defective in strokes. Hence, the repair of the EC–pericyte interaction facilitates vascular maturation and BBB repair.

#### 2.2.5. Microglia/Macrophage

Microglia are the source of the inflammatory cascade activated during a brain injury. Cytokines, including TNF-α, IL-1ββ, and IL-6, are produced in significant amounts by microglia in the brain after an experimental brain injury. The pro-inflammatory phenotype of microglia is associated with tissue destruction, whereas the anti-inflammatory phenotype of microglia facilitates repair and regeneration. Mesenchymal stem cell (MSC) therapy may improve outcomes of an ischemic stroke by inhibiting the activity of the proinflammatory phenotype of microglia and augmenting the activity of the anti-inflammatory phenotype of microglia [85].

Peripheral monocytes infiltrate the ischemic lesion for up to 7 days and then differentiate into macrophages [86]. Inducible NOS (iNOS), expressed in the microglia/macrophages, is detected up to 7 days after MCAO [87]. VEGF-expressing microglia/macrophages may play a role in regeneration through cell–cell interactions [44]. VEGFR1 tyrosine kinase domain-deficient mice are healthy, and angiogenesis is normal in the embryo [88]. In VEGFR1-deficient mice, VEGF-dependent the migration of macrophages is blocked [88], supporting the hypothesis that VEGFR1 also generates a positive signal that stimulates cell migration.

#### 2.2.6. Oligodendrocytes

Pericytes and vascular cells facilitate the proliferation and maturation of OPCs into oligodendrocytes [89,90]. Cerebral ECs, neurons, and astrocytes secrete VEGF [13,91,92], followed by the VEGF-induced migration of OPCs. This effect can be blocked by the VEGFR2 antibody [8]. VEGF-mediated adhesion kinase activation and ROS generation may be important mechanisms involved in OPC migration [8]. Taken together, myelination after CNS injury can be facilitated by repaired cells in the neurovascular unit such as pericytes, ECs, neurons, and astrocytes.

## 3. Neurovascular Repair

Stem cell therapy using MSCs, endothelial progenitor cells (EPCs), and neural stem cells (NSCs) may contribute to stroke recovery as stem cells can secrete a variety of cytokines and growth factors related to angiogenesis, neurogenesis, and synaptogenesis (Figure 3) [93]. Gaseous molecules such as NO and CO also possess regenerative potential demonstrated by boosting stem cell-mediated repair or by stimulating additional regenerative pathways [58,94,95]. Both pathways share common molecular mechanisms associated with VEGF and its related factors (Figure 3).

### 3.1. Stem Cell Therapy

Progress in stem cell biology has significantly contributed to the development of strategies for the treatment of strokes in preclinical studies and has demonstrated clinical potential in stroke treatment. Stem cell therapy may be a promising strategy for the treatment of intractable neurological diseases in the future. Transplanted exogenous stem cell therapy for a brain ischemic stroke may contribute directly to neurovascular regeneration (by compensating for the loss of nerve tissue by the differentiation of nerve and glial cells) as well as indirectly (via secretion of angiogenic and neurogenic factors from the penumbra brain regions). Here, we discuss MSCs, EPCs, and NSCs as potential therapeutic cell types that might be beneficial for the treatment of strokes.

#### 3.1.1. Mesenchymal Stem Cells

MSCs can differentiate into chondrocytes, adipocytes, and osteoblasts, as well as transdifferentiate into ECs, glial cells, and neurons. Owing to their remarkable regeneration potential, MSCs are widely used in current medical research [96]. MSCs secrete a wide range of growth factors, cytokines, chemokines, and extracellular vesicles, thereby contributing to the repair process (i.e., angiogenesis, gliogenesis, and neurogenesis) after an ischemic stroke [97].

The therapeutic potential of MSCs has been demonstrated in ischemic animal models. Human MSCs have been shown to enhance stroke lesion recovery by mediating inflammation and tissue repair through the secretion of trophic factors. Human MSC transplantation into a rat focal ischemia model of transient cerebral artery occlusion revealed decreased accumulation of Iba-1-positive microglia and GFAP-positive astrocytes and the inhibition of pro-inflammatory gene expression in the core and ischemic border zone [87]. MSC therapy may improve outcomes of an ischemic stroke by inhibiting the activity of the proinflammatory phenotype of microglia but augmenting the activity of the anti-inflammatory phenotype of microglia [85]. Human umbilical cord blood-derived MSCs (intravenous injection, 0.25 million cells/animal and 1 million cells/animal) were injected into a rat ischemia–reperfusion stroke model, and experiments were conducted 7 days after reperfusion [98]. The treatment reduced the mRNA and protein levels of metalloproteinases (MMPs) (i.e., MMP-9 and MMP-12) [98].

Apart from the anti-inflammatory responses, MSCs stimulate the regenerative pathway. B10 human MSCs express cytokines and growth factors, including IL-5, fractalkine, insulin-like growth factor-1, glia-derived neurotrophic factor, and VEGF [87]. B10 transplantation also increases the expression of angiogenic factors, such as HIF-1α in the core and border zone of rat ischemic stroke brains [99], which can induce VEGF expression and new vessel formation [99,100]. Allogeneic adipose-derived MSC sheets demonstrated neurological improvement with angiogenesis and neurogenesis in a rat stroke model [101]. Mitochondrial transfer from MSCs to cerebral microvasculature resulted in significant improvement of the mitochondrial activity in injured microvasculature, enhanced angiogenesis, reduced infarct volume, and improved functional recovery following an ischemic stroke [102].

Stromal cell-derived factor-1α (SDF-1α)-transfected MSCs enhance ischemia-mediated new vessel formation as well as angiogenesis in vivo via the VEGF–eNOS axis [103]. IFN-γ-activated MSCs were injected into a rat MCAO model. IFN-γ-activated MSCs demonstrated more potent functional recovery as assessed by the modified neurological severity score and open-field analysis compared to that observed in vehicle-treated animals [104]. IFN-γ-activated MSC-treated stroke-conditioned animals showed a reduction in infarct size, diminished microglial activation, and enhanced recruitment and differentiation of OPCs to myelin-producing oligodendrocytes [104]. Intra-arterial transplantation of 3-dimension (3D) aggregate-derived human MSCs into transient MCAO stroke model mice exhibited increased cell persistence and better therapeutic outcomes compared to that in saline control or 2D human MSC control [105]. The PI3K–Akt signaling pathway was activated by 3D-aggregate-human MSCs [105]. The extracellular regulating kinase 1/2 (ERK) pathway is considered an important regulator in CNS regeneration [58,106]. ERK-overexpressing MSCs were transplanted into stroke model rats, demonstrating the increased proliferation of NSCs and maturation into neurons in the subventricular zone [107]. Glia-like human MSCs (ghMSCs) exhibit better efficacy and enable better protection of the neurons and the brain from ischemia than naïve human MSCs, and insulin-like growth factor binding protein-4 (IGFBP-4) played a critical role in mediating the beneficial effects of ghMSCs in an ischemic stroke [108]. IGFBP-4, hepatocyte growth factor, and VEGF released from ghMSCs may serve as key molecules for enhanced neuronal survival and neurite outgrowth in ischemic CNS injuries [108,109]. Small extracellular vesicles secreted by human-induced pluripotent stem cell-derived MSCs enhance angiogenesis by inhibiting STAT3-dependent autophagy in a rat model of an ischemic stroke [110]. MSC transplantation has also been investigated in humans. Autologous MSC transplantation (intravenous injection, 1 × 10^8^ cells) may improve neurological functions one year after symptom onset in stroke patients [111]. In this study, of the 31 enrolled patients, 16 were administered MSCs. The MSC-treated group showed improvements in motor functioning based on the examination of the National Institutes of Health Stroke Scale score and Fugl-Meyer scores as well as in task-related functional magnetic resonance imaging activity [112]. The transplantation of autologous human MSCs (intravenous injection), cultured in human serum, was performed in 12 stroke patients [113]. In this unblinded study, the mean lesion volume, as assessed by magnetic resonance imaging, was reduced by 420% at one week post-cell infusion [113]. Allogeneic ischemia-tolerant MSCs (intravenous injection, 0.5, 1.0, and 1.5 million cells/kg body weight) were transfused into patients with chronic stroke. Their Barthel index scores increased at 6 months and 12 months post-infusion [114]. Taken together, MSCs may exhibit therapeutic potential by inhibiting excessive inflammation and stimulating the repair pathway.

#### 3.1.2. Endothelial Progenitor Cells (EPCs)

The formation of new blood vessels in the adult brain after a stroke stems from angiogenesis (migration and proliferation of local mature ECs) and the systemic regulation of bone marrow-derived EPCs [93]. CD34 and VEGFR2 double-positive mononuclear cells from peripheral blood are considered as EPCs [115]. EPC mobilization from the bone marrow stroma into the blood circulation is regulated by various enzymes and factors such as eNOS, VEGF, and granulocyte colony-stimulating factor (G-CSF) [43,116,117]. The SDF-1α/C-X-C motif chemokine receptor 4 (CXCR4) pathway plays a key role in the homing of EPCs to the ischemic region [93,118]. The SDF-1α–CXCR4 interaction may recruit not only EPCs, but also MSCs and NSCs to ischemic tissues since the SDF-1–CXCR4 axis modulates survival, proliferation, migration, and differentiation of MSCs and NSCs [119,120].

Ex vivo expanded EPCs (intravenous injection, 1 × 10^6^ cells) were injected into mice after 1 h following induction of transient MCAO [121]. EPC transplantation significantly reduced ischemic infarct volume and induced angiogenesis in the ischemic penumbra after MCAO compared to that observed in control mice in vivo, and a CXCR4 antagonist blocked SDF-1-mediated EPC migration in vitro [121]. Moreover, SDF-1 upregulates VEGF expression and eNOS activity via cellular communication [103].

The interplay between eNOS and BDNF may be involved in EPC-mediated angiogenesis, neurogenesis, and axonal growth after an ischemic stroke [122]. Conditioned media derived from EPC culture was administered to mice 1 d after MCAO. A significant increase in capillary density was observed in the ischemic penumbra, consequently improving forelimb strength [123]. The expression of multiple growth factors, cytokines, and proteases has been demonstrated in the EPC secretome, showing enhanced endothelial and OPC proliferation and maturation [90]. Angiogenin, a HIF-1α target gene, may be a key factor since pharmacological blockade of angiogenin signaling negates the positive effects of the EPC secretome [90,124]. Under in vivo conditions, treatment with the EPC secretome increases vascular density, myelin, and mature oligodendrocytes in the white matter and rescues cognitive function in a mouse model of hypoperfusion [90]. Hypoxic preconditioning via overexpression of HIF-1α, SDF-1α, VEGFR2, or VEGF may facilitate EPC functions such as angiogenesis and neurogenesis [125,126].

Autologous CD34-positive stem/progenitor cells derived from the bone marrow of human subjects were administered intra-arterially via catheter angiography within 7 days of the onset of a severe ischemic stroke [127]. In this study, administration of CD34-positive stem cells resulted in reduced lesion volume and hence, rescued patients with an acute ischemic stroke during a 6-month follow-up period [127]. Thus, the application of EPCs in an ischemic stroke may be helpful.

#### 3.1.3. Neural Stem Cells (NSCs)

IL-17A shows two distinct peaks of expression in the ischemic hemisphere: the first peak observed within 3 days and the second on day 28 after a stroke. Astrocytes are a major cellular source of IL-17A, which maintains and augments subventricular zone (SVZ) neuronal precursor cell survival, neuronal differentiation, synaptogenesis, and functional recovery after a stroke [128]. In this study, the p38 mitogen-activated protein kinase–calpain 1 signaling pathway was involved in IL-17A-mediated neurogenesis [128].

NAMPT is a rate-limiting enzyme involved in the biosynthesis of nicotinamide adenine dinucleotide (NAD) in mammals; this putative therapeutic agent for combating stroke is highly expressed in neurons, EPCs, and NSCs, with lower expression in glial cells [129,130]. It plays key roles in defense mechanisms, metabolic homeostasis, and neuronal survival [129]. NAD replenishment in neurons either before or after oxygen-glucose deprivation reduces cell death and DNA damage [131]. Neuronal survival due to NAMPT overexpression was blocked in AMPKα2−/− neurons through the SIRT1–AMPK axis in a rat model of an ischemic stroke [132]. The role of NAMPT has been demonstrated in neurovascular repair during the chronic phase. NAMPT promotes angiogenesis, neovascularization, and neurite outgrowth as well as increases the levels of regenerative factors such as BDNF and VEGF [129,133,134,135].

### 3.2. Gaseous Biomolecules

Reduced O_2_ availability (i.e., hypoxia) and energy substrates (e.g., glucose) appear to represent the critical stimulus that evokes an adaptive response to ischemia, principally through HIF-dependent production of multiple angiogenic cytokines and growth factors, including VEGF, angiogenin, angiopoietins, placental growth factor, PDGF-B, stem cell factor, and SDF-1, which stimulate angiogenesis, the process of new blood vessel formation from pre-existing ones [27,124,136,137,138,139].

The endogenous repair ability of gaseous biomolecules such as NO and CO can be promoted after a stroke. Nitric oxide (NO) and CO are endogenous gases produced by NOS and HO, which diffuse freely between cells, consequently amplifying signaling pathways involved in neurogenic and angiogenic functions [42,137]. NO can dilute the cerebral vasculature and enhance cerebral blood flow. CO improves damaged vasculature by inducing angiogenesis and neovascularization, partly by interacting with the NOS signaling pathway [94]. Moderate levels of NO and CO induces HIF-1α-mediated VEGF expression [75,140] and suppress its expression in severe hypoxia [141].

#### 3.2.1. NO

NO is produced by the reaction of l-arginine with NOS isoforms. Two constitutive isoforms, (eNOS and neuronal NOS [nNOS]) via Ca^2+^ entry, and iNOS are enzymes that are expressed in a highly cell type-specific manner. Acting as an intercellular signal, the nNOS–NO axis can trigger neurogenesis in mouse brain neural progenitor cells. BDNF upregulates nNOS protein levels, which can induce the maturation of neurons from neural progenitor cells [142].

VEGF–VEGFR2–PI3K–Akt axis is an important regulator of cellular survival, cell motility, and NO production [143,144]. Activation of AMPK, a crucial cellular energy sensor, can also stimulate eNOS by phosphorylating it at Ser^1179^, suggesting crosstalk between cellular metabolism and vascular tone. By using flow channels with cultured ECs, AMPKα Thr^172^ phosphorylation can be increased with changes in flow rate or pulsatility [145].

eNOS exhibits neuroprotective properties against ischemic strokes [43]. Moderate NO gas inhalation in mice with transient focal ischemia reduced infarct volume to 10 ppm for 24 h, and to 20, 40, and 60 ppm for 8 and 16 h following NO inhalation [146]. NO inhalation improves penumbral blood flow and neurological outcomes in a mice ischemia induced by transient MCAO [147] and in a rat model of focal cerebral ischemia [63].

An HO inducer or carbon monoxide-releasing molecule 2 (CORM-2) restores TNF-α-induced downregulation of the expression of eNOS–NO by inhibiting NF-κB-responsive miR-155-5p expression in HUVECs [42]. CO can reduce the production of ROS, consequently reducing the synthesis of peroxynitrite [148]. Application of CORM-2 in BV2 microglial cells prevents the production of NO upon lipopolysaccharide (LPS) stimulation [149]. Therefore, CO may reduce LPS-mediated NO production in activated glial cells and stimulate NO production in vascular cells. In addition, CO may facilitate repair and regeneration by activating the nNOS–NO pathway in neuronal cells [95]. The interplay between CO and NO leads to vascular dilation, angiogenesis, and neurogenesis.

The nNOS–NO axis is activated in the neurons and NSCs when mice are injected with CORM-3 after an ischemic CNS injury [95]. CO may exert neurogenic effects by stimulating the HO-1 pathway, consequently activating the nNOS–NO axis. The HO metabolite, bilirubin, stimulates ERK1/2 phosphorylation, CREB phosphorylation, and nNOS–NO production in the absence of exogenous growth factors in PC12 cells. This effect is blocked by an extracellular Ca^2+^ chelator [150]. NMDAR, a critical receptor for hippocampal long-term potentiation and spatial learning, is S-nitrosylated by NO [150]. Thus, the crosstalk between HO/CO and NOS/NO may induce diverse signaling pathways leading to neurogenesis, long-term potentiation, learning, and memory.

#### 3.2.2. Carbon Monoxide (CO)

HO-1 and HO-2 are essential enzymes in heme catabolism that cleave heme to carbon monoxide (CO), biliverdin (which is rapidly converted to bilirubin), and Fe^2+^. In this step, O_2_ is required as the reaction substrate [136]. HO-2 is constitutively expressed in neurons where it functions as an intrinsic protector [151]. The expression of HO-1 is strongly induced in various cells in response to hypoxia and stress, which promotes neuroprotection and angiogenesis in the ischemic milieu [152]. CO induces the expression of HO-1 and plays important roles in neurotransmission, neurogenesis, mitochondrial biogenesis, and blood circulation in the brain [153,154,155]. CO generated either by exogenous delivery or by HO activity is fundamentally involved in regulating mitochondria-mediated redox cascades for adaptive gene expression and improving blood circulation (i.e., O_2_ delivery) via neovascularization, leading to the regulation of mitochondrial energy metabolism [137]. CO can be delivered in a pharmacologically active form as a CO-releasing molecule, CORM [156]. The biological effects of CO are largely dependent on the HO activity [137].

HO-derived CO promotes angiogenesis and neovascularization by regulating pro-angiogenic VEGF expression [157,158,159,160]. CO can also upregulate HO-1 expression, and the HO-1/CO circuit may interact with the NOS/NO pathway [94]. CORM-2 can stimulate the eNOS–NO axis through inositol triphosphate receptor-mediated intracellular Ca^2+^ release, PI3K-Akt phosphorylation, and eNOS dimerization in HUVECs [161]. CORM-2 prevents TNF-α-induced eNOS downregulation by inhibiting NF-κB-responsive miR-155-5p biogenesis [42]. Inflammatory responses after ischemic stroke are diminished by CORM-3, demonstrated by examining the levels of TNF-α and IL-1ββ [162]. CO-mediated ROS/RNS inhibition may protect the BBB from acute neuroinflammatory diseases [95,162].

Signaling pathways activated by CORM-2 include the PI3K-HIF-1α–VEGF pathway [75]. In a mouse model of ischemia–reperfusion injury, HO-1 is expressed in astrocytes in the penumbra region [19]. Transient HO-1 activation may be beneficial for regeneration during an acute ischemic injury [95,156,163,164,165,166]. HO-1-derived CO and bilirubin activate LTCC and mediate Ca^2^^+^/CaMKKββ-mediated activation of AMPKα, AMPKα-dependent HO-1 induction, and the consequent stabilization of HIF-1α in a PHD2-dependent manner [49]. The effects of CO on regeneration are associated with VEGF production [19]. Recently, the neuroprotective and regenerative effects of CORM-3 have been demonstrated in a stroke. CORM-3 injection reduced infarct volume and increased the expression of mature neuronal markers such as neuronal nuclear antigen and microtubule-associated protein 2 compared to that in saline-treated mice [162]. CO-mediated VEGF upregulation does not disrupt BBB, instead, CO protects the BBB from ischemic injury [95,162]. Other CO-mediated protective factors may mitigate VEGF-mediated vascular permeability, or CO reduces excessive VEGF production. Taken together, CO reduces VEGF-mediated disruption of BBB and facilitates VEGF-induced regeneration after an ischemic stroke.

## 4. Discussions/Conclusions

The present review discusses the repair potential of various exogenous stem cells and gaseous molecules with respect to neurovascular protection and regeneration after an ischemic stroke. Ischemic stroke is a complex disease with multiple underlying pathways. To address this multiplicity, therapeutic agents may target more than one pathway, such as anti-inflammation, neuroprotection, and neurovascular regeneration involved in angiogenesis, neurogenesis, and gliogenesis. Commonly shared key molecular mechanisms in the neurovascular unit repair are associated with the VEGF and its related factors.

A neurotrophic supportive environment may contribute to neurovascular regeneration and the formation of functional neural circuits upon an ischemic stroke. Exogenous stem cells and gaseous molecules have been used to treat ischemic strokes. This strategy includes enhancing VEGF-mediated regeneration and the ability of VEGF-related factors to amplify the cellular network in the CNS. The crosstalk between cells and the regenerative ability of these cell-derived factors boosts renewal potential by activating the functions of endogenous NSCs (Figure 4).

Nevertheless, current studies examining the repair following an ischemic stroke have limited applicability for humans, possibly owing to the disconnection between animal models employed by the laboratory and actual disease states observed in humans [167,168]. In addition to developing proper animal models for representing an ischemic stroke, the administration of a single drug to ischemic stroke patients would not be sufficient to repair the entire neurovascular unit. Therefore, the administration of a cocktail of therapeutic drugs, discussed in this review, with multiple regenerative mechanisms may be beneficial.

## Figures and Tables

**Figure 1 ijms-22-08543-f001:**
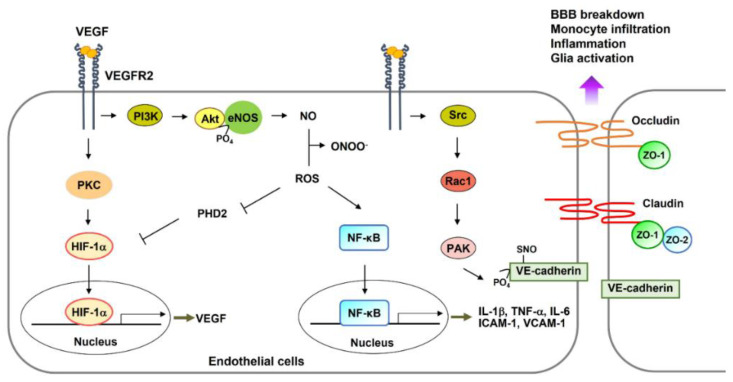
Following an acute ischemic stroke, VEGF, whose expression is enhanced, binds with VEGFR2 and triggers various signaling pathways in endothelial cells. For instance, the VEGFR2–PKC pathway may activate the transcription factor of VEGF and HIF-1α. Next, the PI3K–Akt pathway phosphorylates eNOS and activates eNOS enzymatic functions leading to NO production. Oxidative stress mediates ROS production. ROS reacts with NO to synthesize peroxynitrite (ONOO^−^). ROS signaling inhibits PHD2 activity, leading to HIF-1a protein stabilization and activation of NF-κB protein, consequently upregulating expression of downstream genes related to NF-κB, such as IL-1β, TNF-α, IL-6, ICAM-1, and VCAM-1. ONOO^−^ modifies VE-cadherin protein by nitrosylation of serine residues, resulting in the detachment of VE-cadherin from endothelial cells. Finally, the VEGFR2-mediated Src–Rac1–PAK pathway phosphorylates VE-cadherin, resulting in dissociation with the adherence junction between endothelial cells. Oxidative stress-mediated reduction in the expression of tight junction proteins also initiates BBB breakdown and monocyte infiltration. HIF-1α- and NF-κB-mediated gene expression facilitates BBB breakdown, inflammatory responses, and glial scarring. Abbreviations: VEGF, vascular endothelial growth factor; VEGFR, VEGF receptor; PI3K, phosphatidylinositol 3-kinase; PKC, protein kinase C; NO, nitric oxide; eNOS, endothelial nitric oxide synthase; HIF-1α, hypoxia-induced factor-1α; NF-κB, nuclear factor κ-light-chain-enhancer of activated B cells; PAK, p21-activated kinase; IL, interleukin; TNF-α, tumor necrosis factor α; VE-cadherin, vascular endothelial cadherin; ICAM, intercellular adhesion molecule; VCAM, vascular cell adhesion molecule; ZO, zonula occludens; ROS, reactive oxygen species; PHD, prolyl-4-hydroxylase domain; BBB, blood–brain barrier.

**Figure 2 ijms-22-08543-f002:**
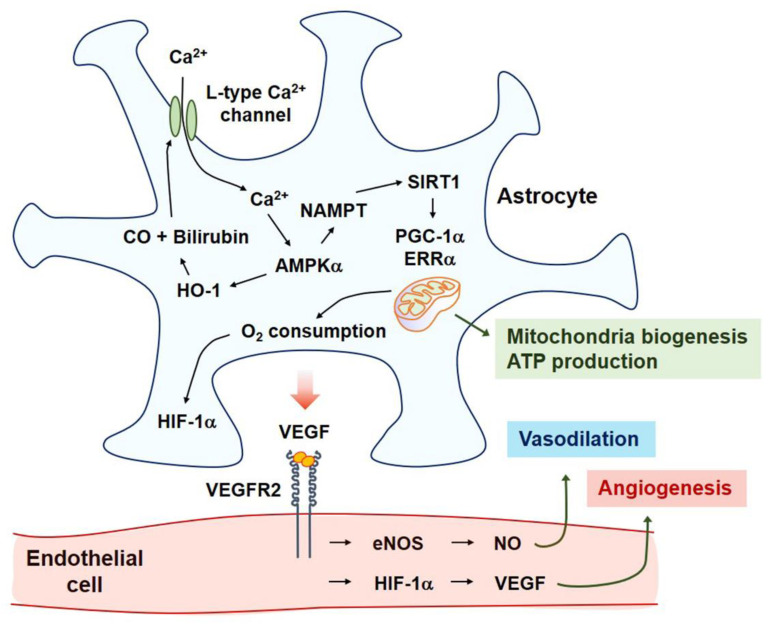
After the ischemia-reperfusion injury, HO-1 is expressed in the astrocytes of the penumbra of the ischemic brain. CO and bilirubin can be produced via enzymatic activity of HO-1, which then activates the L-type Ca^2+^ channel, leading to the entry of extracellular Ca^2+^. Ca^2+^-mediated AMPKα-HO-1 circuit results in NAMPT gene expression and consequent SIRT1 activation. Deacetylation of PGC-1α by SIRT1 activates *ERRα* expression. PGC-1α-ERRα axis induces mitochondrial biogenesis, ATP production, and increased O_2_ consumption. Reduction of O_2_ inhibits PHD2 activity and then stabilizes HIF-1α protein. Both the PGC-1α-ERRα pathway and HIF-1α pathway induce the expression and secretion of VEGF. VEGF dimerization and binding to VEGFR2 triggers the eNOS–NO axis and HIF-1α–VEGF pathway in endothelial cells, leading to vasodilation and angiogenesis, respectively. Abbreviations: CO, carbon monoxide; HO-1, heme oxygenase-1; AMPKα, AMP-activated protein kinase α; PGC-1α, peroxisome proliferator-activated receptor γ-coactivator-1α; PHD, prolyl-4-hydroxylase domain; ERRα, estrogen-related receptor α; SIRT1, sirtuin 1; NAMPT, nicotinamide phosphoribosyltransferase.

**Figure 3 ijms-22-08543-f003:**
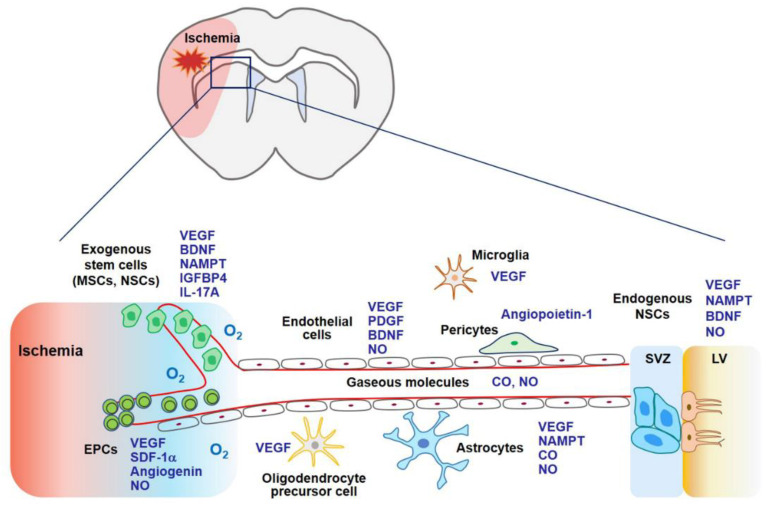
In the later phase of an ischemic stroke, VEGF expression may facilitate tissue regeneration. Exogenous stem cells (i.e., MSCs, EPCs, and NSCs) and gaseous molecules (i.e., NO and CO) increase the expression of VEGF and its related factors, which are involved in angiogenesis, neurogenesis, and synaptogenesis. Additionally, these factors enhance the ability of endogenous NSCs to proliferate and differentiate into mature neurons, astrocytes, and oligodendrocytes. Collectively, regeneration through stem cells and gaseous molecules contributes to repair after an ischemic stroke. Abbreviations: MSCs, mesenchymal stem cells; NSCs, neural stem cells; EPCs, endothelial precursor cells; BDNF, brain-derived neurotrophic factor; IGFBP4, insulin-like growth factor binding protein-4; PDGF, platelet-derived growth factor; SDF-1α, stromal cell-derived factor-1α; SVZ, subventricular zone; LV, lateral ventricle.

**Figure 4 ijms-22-08543-f004:**
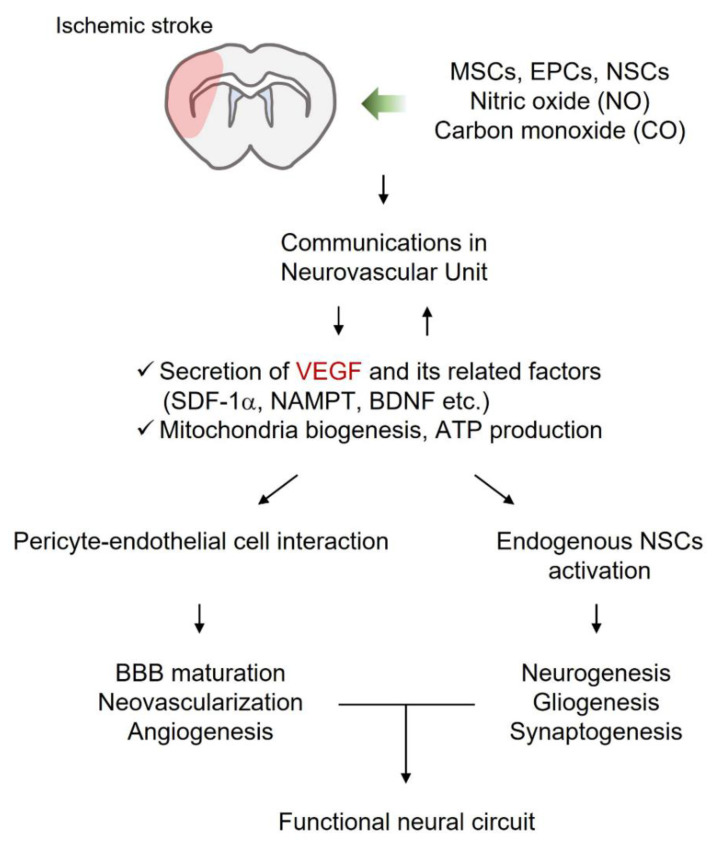
Neuronal growth-promoting environment supported by exogenous stem cells and gaseous molecules may contribute to repair upon ischemic stroke. Cellular and molecular crosstalk in the neurovascular unit leads to BBB maturation, angiogenesis, neurogenesis, gliogenesis, and synaptogenesis. Collectively, damaged tissue may be repaired through these regenerative pathways and form new functional neural circuits. Abbreviations: BBB, Blood–Brain Barrier; BDNF, brain-derived neurotrophic factor; EPCs, endothelial precursor cells; MSCs, mesenchymal stem cells; NAMPT, nicotinamide phosphoribosyltransferase; NSCs, neural stem cells; SDF-1, stromal cell-derived factor-1; VEGF, vascular endothelial growth factor.

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
