# Peer review of "Repair Mechanisms of the Neurovascular Unit after Ischemic Stroke with a Focus on VEGF"

_ijms, 2021, doi:10.3390/ijms22168543_

Round 1
Reviewer 1 Report
Scientifically sound review, very well written and supported, as it includes appropriate and adequate references. The work is well organized and comprehensively described and not scientifically misleading.
The only remark from my side is that there is no separate discussion and conclusion section, which potentially weakens the contribution of your review to the field. I have noticed that there is a section under the term ‘Perspectives’ which attempts to substitute the conclusion section, but this is disproportionately small, in comparison with the rest of the manuscript. I would expect a more extended discussion section, explaining in more details the sources of the data that are presented and if there any experimental models that were used in order to study stroke in animals.
I would also like to mention what Youdim wrote in 2001: ‘The failure to induce neuroprotection in the clinic versus in the laboratory with currently available drugs would suggest that either the animal models we are employing are not truly representative of the disease state’. I would like to state that you should mention that the lack of translation between the animal work and clinical benefits lies in how we use the models and how we apply this knowledge to design of clinical trials. This means that we should underlie the fact that, although all of the presented potential therapeutics may have therapeutic implications in animal models of experimental stroke, not all of them are expected to have similar beneficial effects in clinical trials.
Author Response
Response: Thank you for your comment. We extended our discussion/conclusion by replacing ‘Perspectives’ and by adding Youdim’s suggestion (line 597-616).
- Discussion/Conclusion
Ischemic stroke is a complex disease with multiple underlying pathways. To address this multiplicity, therapeutic agents may target more than one pathway, such as anti-inflammation, neuroprotection, and neurovascular regeneration involved in angiogenesis, neurogenesis, and gliogenesis.
A neurotrophic supportive environment may contribute to neurovascular regeneration and the formation of functional neural circuits upon ischemic stroke. Exogenous stem cells and gaseous molecules have been used to treat ischemic stroke. This strategy includes enhancing VEGF-mediated regeneration and the ability of VEGF-related factors to amplify the cellular network in the CNS. The crosstalk between cells and the regenerative ability of these cell-derived factors boosts renewal potential by activating the functions of endogenous NSCs (Figure 4).
Nevertheless, current studies examining the repair following ischemic stroke have limited applicability for humans, possibly owing to the disconnection between animal models employed by the laboratory and actual disease states observed in humans [167, 168]. In addition to developing proper animal models for representing ischemic stroke, administration of a single drug to ischemic stroke patients would not be sufficient to repair the entire neurovascular unit. Therefore, the administration of a cocktail of therapeutic drugs discussed in this review with multiple regenerative mechanisms may be beneficial.

Reviewer 2 Report
The authors reviewed different potentially repairing pathways after stroke with a common final step including VEGF. There are already many reviews on pathogenic mechanisms of stroke and potential therapeutic avenues, so the focus selected should provide a really novel approach to the topic to make it appealing. This is not fully achieved.
General ideas are well explained, and the overlay of the text is sound. However, the manuscript is not easy to read. Too many observations/ideas are included not always following a logical structure, going out of the topic of the subheading. The authors need to re-order the text avoiding mixing concepts and observations. At times, the details of previous observations contribute to make the text a bit confusing. The text would benefit from a more synthetic approach to present knowledge.
Specific comments.
Title: is somewhat misleading as the focus is not only on VEGF, but also in gaseous molecules and stem cells.
Abstract: the neurovascular unit contains neurons also. Oligodendrocytes/stem cells are not usually considered part of this functional organization. The text exemplifies the continued mixing of concepts/ideas. The authors start describing the repair actions of stem cells and gaseous mediators and end up commenting that they review endogenous stem cells and endothelial cells and their repair capacity.
A general introductory statement of pathogenic mechanisms in stroke would help position the interest of VEGF within the myriad mechanisms described in the pathogenic cascade. Why they selected it?
Figure 1 requires further details to put it into context (potential sources of VEGF, include pathways commented in the legend…etc)
One example of discontinued structuring of the text: paragraph 463-487. There is a constant mix of observations around different topics.
Minor: language should be carefully edited
Author Response
Reviewer #2
Comment #1: The authors reviewed different potentially repairing pathways after stroke with a common final step including VEGF. There are already many reviews on pathogenic mechanisms of stroke and potential therapeutic avenues, so the focus selected should provide a really novel approach to the topic to make it appealing. This is not fully achieved.
Response #1: VEGF and its related factors are involved in entire neurovascular repair by regulating multifaceted pathways stimulating endothelial survival, angiogenesis, neurogenesis, gliogenesis, and synaptic functions [3-8]. Thus, we mentioned that ‘In this review, we have focused on VEGF-A and refer to it as VEGF because VEGF acts on entire neurovascular cells such as endothelial cells, neurons, and glia [3-8]. All these mechanisms involving stem cells and gaseous molecules partly share common molecular pathways and signaling molecules such as VEGF and its related factors.’ (line 54-58).
References
- Cao, L.; Jiao, X.; Zuzga, D. S.; Liu, Y.; Fong, D. M.; Young, D.; During, M. J., VEGF links hippocampal activity with neurogenesis, learning and memory. Nat Genet 2004, 36, (8), 827-35.
- De Rossi, P.; Harde, E.; Dupuis, J. P.; Martin, L.; Chounlamountri, N.; Bardin, M.; Watrin, C.; Benetollo, C.; Pernet-Gallay, K.; Luhmann, H. J.; Honnorat, J.; Malleret, G.; Groc, L.; Acker-Palmer, A.; Salin, P. A.; Meissirel, C., A critical role for VEGF and VEGFR2 in NMDA receptor synaptic function and fear-related behavior. Mol Psychiatry 2016, 21, (12), 1768-1780.
- Domigan, C. K.; Warren, C. M.; Antanesian, V.; Happel, K.; Ziyad, S.; Lee, S.; Krall, A.; Duan, L.; Torres-Collado, A. X.; Castellani, L. W.; Elashoff, D.; Christofk, H. R.; van der Bliek, A. M.; Potente, M.; Iruela-Arispe, M. L., Autocrine VEGF maintains endothelial survival through regulation of metabolism and autophagy. J Cell Sci 2015, 128, (12), 2236-48.
- During, M. J.; Cao, L., VEGF, a mediator of the effect of experience on hippocampal neurogenesis. Curr Alzheimer Res 2006, 3, (1), 29-33.
- Eilken, H. M.; Dieguez-Hurtado, R.; Schmidt, I.; Nakayama, M.; Jeong, H. W.; Arf, H.; Adams, S.; Ferrara, N.; Adams, R. H., Pericytes regulate VEGF-induced endothelial sprouting through VEGFR1. Nat Commun 2017, 8, (1), 1574.
- Hayakawa, K.; Pham, L. D.; Som, A. T.; Lee, B. J.; Guo, S.; Lo, E. H.; Arai, K., Vascular endothelial growth factor regulates the migration of oligodendrocyte precursor cells. J Neurosci 2011, 31, (29), 10666-70.
Comment #2: General ideas are well explained, and the overlay of the text is sound. However, the manuscript is not easy to read. Too many observations/ideas are included not always following a logical structure, going out of the topic of the subheading. The authors need to re-order the text avoiding mixing concepts and observations. At times, the details of previous observations contribute to make the text a bit confusing. The text would benefit from a more synthetic approach to present knowledge.
Response #2: Per your suggestion, we organized the text in a more synthetic approach by removing some papers.
Specific comments.
Comment #3: Title: is somewhat misleading as the focus is not only on VEGF, but also in gaseous molecules and stem cells.
Response #3: Molecular mechanisms and factors related with Stroke are diverse. However, all these mechanisms partly share common molecular pathways and signaling molecules such as VEGF and its related factors even in gaseous molecules and stem cells. Therefore, we our title is changed into “Repair Mechanisms of the Neurovascular Unit After Ischemic Stroke with Focus on VEGF”.
Comment #4: Abstract: the neurovascular unit contains neurons also. Oligodendrocytes/stem cells are not usually considered part of this functional organization. The text exemplifies the continued mixing of concepts/ideas. The authors start describing the repair actions of stem cells and gaseous mediators and end up commenting that they review endogenous stem cells and endothelial cells and their repair capacity.
Response #4: We apologize for suboptimal explanation. We added neurons in neurovascular unit (line 12): In the CNS, neurovascular units, including neurons, endothelial cells---
Additionally, we added neurons/ NSCs in section 2.2. Regeneration (line 210-235): In this section, we discuss VEGF-mediated signaling pathways with respect to neurovascular components, such as neurons, NSCs, ECs, pericytes, astrocytes, microglia, and oligodendrocytes.
2.2.1. Neurons/NSCs
Accumulating evidence indicates that the VEGF-VEGFR2 axis mediates key neurobiological processes involved in neurogenesis, hippocampal plasticity, learning, and memory [3, 20]. Co-activation of NMDA receptor (NMDAR) and VEGFR2 in hippocampal pyramidal neurons triggers synaptogenesis and promotes synaptic targeting of NMDAR and VEGFR2 [4]. Overexpression of VEGF in vivo by intracerebral administration, gene transfer, or in conditional transgenic animal models demonstrates VEGFR2-mediated increase in adult neurogenesis and improves hippocampal-dependent cognition [59]. Moreover, VEGFR2 is required for the beneficial effects of anti-depressant therapies on fear-related behavior [4]. In contrast, depletion of endogenous VEGF via small hairpin RNA silencing or inducible expression of a VEGF trap leads to altered hippocampal neurogenesis in response to an enriched environment [3, 6] or selective deficits in memory [59]. Both VEGF and VEGFR2 are expressed in the pyramidal cell layer and proximal apical dendrites in the cornu ammonis 1 (CA1) and CA3 regions of the hippocampi in postnatal day 15 mice [4], indicating that hippocampal pyramidal cells can respond to endogenous VEGF. Treatment with an inhibitor that specifically inhibits the tyrosine kinase activity of VEGFR2 diminishes the amplifying effect of VEGF [4]. NSCs, genetically modified by deleting the VEGFR2 gene, inhibit synaptogenesis, which is identified by detecting the expression of presynaptic vesicle marker synapsin-1 and/or the postsynaptic density protein-95 [4]. Conditional KO of VEGFR2 via Nestin-cre system triggers selective VEGFR2 deletion in neural cells. Heterozygote tissue sections exhibit increased NMDA type of glutamate receptor (GluNRs)-mediated synaptic responses when VEGF is administered, whereas VEGF fails to induce any significant change in GluNR excitatory postsynaptic currents in VEGFR2 conditional KO slices [4]. Therefore, NSC-derived VEGFR2-VEGF signaling may trigger regeneration processes such as neurogenesis and synaptogenesis, leading to enhanced long-term memory.
This paragraph was moved from 463-487 (original version of text) for more organized form of text.
Comment #5: A general introductory statement of pathogenic mechanisms in stroke would help position the interest of VEGF within the myriad mechanisms described in the pathogenic cascade. Why they selected it?
Response #5: When we searched keywords in PUBMED (www.ncbi.nlm.nih.gov) on July 20, 2021, there are following reports:
- Stroke, VEGF = 1,452 papers
- Stroke, IL-17A = 105 papers
- Stroke, HGF = 86 papers
- Stroke, Nampt = 22 papers
Additionally, VEGF and its related factors are involved in entire neurovascular repair by regulating multifaceted pathways stimulating endothelial survival, angiogenesis, neurogenesis, gliogenesis, and synaptic functions [3-8]. Thus, we mentioned that ‘In this review, we have focused on VEGF-A and refer to it as VEGF because VEGF acts on entire neurovascular cells such as endothelial cells, neurons, and glia [3-8]. All these mechanisms involving stem cells and gaseous molecules partly share common molecular pathways and signaling molecules such as VEGF and its related factors.’ (line 54-58).
Taken together, we selected VEGF as an important molecule.
Comment #6: Figure 1 requires further details to put it into context (potential sources of VEGF, include pathways commented in the legend…etc)
Response #6: We put (Figure 1) into context (line 99-101): Hypoxic astrocyte-derived VEGF binds to VEGFR2 expressed on human retina microvascular ECs, leading to HIF-1a gene expression and consequent VEGF upregulation [30] (Figure 1).
We put (Figure 1) into context (line 144-148): Peripheral monocytes attach to the damaged brain ECs expressing intercellular adhesion molecule-1 (ICAM-1) and vascular cell adhesion molecule-1 (VCAM-1), which are downstream factors of kappa-light-chain-enhancer of activated B cells (NF-κB) observed during acute CNS inflammation [41] (Figure 1).
We put (Figure 1) into context (line 255-256): Activation of VEGFR2 in ECs via binding with a strong agonist, VEGF, results in Akt-dependent phosphorylation of eNOS as well as NO production [62] (Figure 1).
Comment #7: One example of discontinued structuring of the text: paragraph 463-487. There is a constant mix of observations around different topics.
Response #7: Per your suggestion, we moved this paragraph into section 2.2. regeneration (line 210-235) for more structural format.
Comment #8: Minor: language should be carefully edited
Response #8: We did our best at having our manuscript carefully edited.
